# Topical application of calcitonin gene-related peptide as a regenerative, antifibrotic, and immunomodulatory therapy for corneal injury
Asmaa A. Zidan, Shuyan Zhu, Elsayed Elbasiony, Sheyda Najafi, Zhirong Lin, Rohan Bir Singh, Amirreza Naderi & Jia Yin ✉

Calcitonin gene-related peptide (CGRP) is a multifunctional neuropeptide abundantly expressed by corneal nerves. Using a murine model of corneal mechanical injury, we found CGRP levels in the cornea significantly reduced after injury. Topical application of CGRP as an eye drop accelerates corneal epithelial wound closure, reduces corneal opacification, and prevents corneal edema after injury in vivo. CGRP promotes corneal epithelial cell migration, proliferation, and the secretion of laminin. It reduces TGF-β1 signaling and prevents TGF-β1-mediated stromal fibroblast activation and tissue fibrosis. CGRP preserves corneal endothelial cell density, morphology, and pump function, thus reducing corneal edema. Lastly, CGRP reduces neutrophil infiltration, macrophage maturation, and the production of inflammatory cytokines in the cornea. Taken together, our results show that corneal nerve-derived CGRP plays a cytoprotective, pro-regenerative, anti-fibrotic, and anti-inflammatory role in corneal wound healing. In addition, our results highlight the critical role of sensory nerves in ocular surface homeostasis and injury repair.

Corneal opacity is a leading cause of visual impairment, affecting more than 5.5 million people worldwide[1]. Although corneal opacity is associated with several underlying etiologies, trauma accounts for nearly one-third of all cases[2]. Mild corneal opacity can be treated with topical corticosteroids, which carry potential side effects including increased intraocular pressure, risks of infection, and cataract formation[3]. Current therapeutic modalities for moderate to severe corneal opacity are limited to corneal transplantation with issues such as limited tissue supply[4], surgical complications, and graft failure[5]. Therefore, there is an urgent need to develop alternative and novel therapies that can efficaciously prevent corneal opacification and promote wound healing and tissue regeneration.

Post-traumatic corneal healing is a complex orchestrated interplay between the corneal cells, extracellular matrix, and inflammatory cells[6]. While superficial corneal injuries limited to the epithelium usually heal without a scar, disruption of the epithelial basement membrane (EBM) induces stromal fibrosis[7]. The disruption of EBM allows growth factors such as transforming growth factor beta-1 (TGF-β1) and inflammatory cytokines secreted from apoptotic epithelial cells to leak into the stroma, inducing the differentiation of stromal keratocytes to myofibroblasts and recruiting inflammatory cells to the injury site[8]. This "healing" process primarily functions to clear and replace the damaged and apoptotic corneal epithelial and stromal cells. However, uncontrolled myofibroblast activation and persistent inflammation eventually lead to corneal opacity[9]. Furthermore, uncontrolled inflammation leads to corneal endothelial cell dysfunction and apoptosis, aggravating corneal edema and opacity[10].

The cornea is the most densely innervated tissue in the human body with sensory nerves derived from the ophthalmic branch of the trigeminal nerve (cranial nerve V1). In addition to sensory function, corneal nerves regulate blink reflex, and tear production, and contribute to the maintenance of stem cells and ocular immunity[11,12]. The role of corneal nerves in wound healing has been reported[13,14]. A recent study highlighted the critical function of sensory nerves in supporting epithelial stem cells in a murine model of ocular injury[15]. Additionally, it was reported that sensory nerve degeneration and dysfunction in diabetic and neurotrophic keratopathy can lead to recurrent ulceration, delayed wound healing, and corneal opacification[16]. Corneal nerves secrete neuropeptides that carry out

Schepens Eye Research Institute of Massachusetts Eye and Ear, Department of Ophthalmology, Harvard Medical School, Boston, MA, USA.
✉e-mail: jia_yin@meei.harvard.edu

regulatory functions at the ocular surface[17]. One of these neuropeptides is calcitonin gene-related peptide (CGRP), which is expressed by two-thirds of corneal nerves[18].

CGRP is a 37-amino acid long peptide and highly expressed in the central and peripheral nervous systems[19]. It has an array of physiological functions through binding to calcitonin receptor-like (CLR) and receptor activity-modifying (RAMPs) receptors[20–22]. While corneal nerves express high levels of CGRP, its receptors have been found in the corneal epithelium, stroma, and endothelium[17]. Upon binding to CLR-RAMP1, CGRP activates adenylyl cyclase, increases intracellular cAMP concentration, and subsequently activates protein kinase A, C and MAP kinase which regulates epithelial migration and proliferation[23]. CGRP has been demonstrated to accelerate wound healing and inhibit tissue fibrosis in the skin and bronchial epithelium[23–25]. The effect of CGRP on inflammation is tissue- and context-dependent[26–31].

It has been reported that CGRP levels in the tear film are changed after corneal surgeries and trauma[32,33], but its level in the cornea after the injury has not been examined. More importantly, while CGRP has been shown to promote corneal re-epithelialization in vitro in a single study[34], the efficacy of topically applied CGRP as an eyedrop in treating corneal stromal injury and promoting wound healing and regeneration in vivo remains unknown. Herein, we use a well-established murine model of corneal epithelial-stromal injury and inflammation to determine the cytoprotective, regenerative, and immunosuppressive functions of CGRP.

## Results

### Mechanical injury causes a decrease in CGRP levels in the cornea

A 2-mm-wide mechanical injury was induced at the center of the cornea by removing the epithelium and superficial stroma (Fig. 1a). We observed that the CGRP levels in the cornea were significantly lower on days 1, 3, and 7 post-injury (Fig. 1b). We also assessed levels of CGRP receptors CLR, RAMP1, and RAMP2 in the cornea and found a significant upregulation in RAMP1 gene expression level, which peaked at day 3 post-injury and returned to pre-injury level by day 7 (Fig. 1c). The expression levels of CLR and RAMP2 were unchanged post-injury.

### Topical application of CGRP accelerates corneal epithelial wound closure and reduces corneal opacity

We applied CGRP (5 µl of a 50 µM CGRP solution diluted in PBS as an eye drop) topically three times daily immediately after injury for 14 days (Fig. 2a). The control animals were treated with an equal volume of PBS eye drops. Corneal epithelial wound closure was highlighted with fluorescein staining (Fig. 2b, c). The remaining wound area post-injury decreased more rapidly in CGRP-treated mice, compared to the controls. Slit lamp photography showed gradual opacification of the cornea after injury in the control group, whereas CGRP treatment led to a significant decrease in corneal opacity (opacity score PBS vs CGRP) on day 7 ($3.1 \pm 0.3$ vs $1.9 \pm 0.3$,

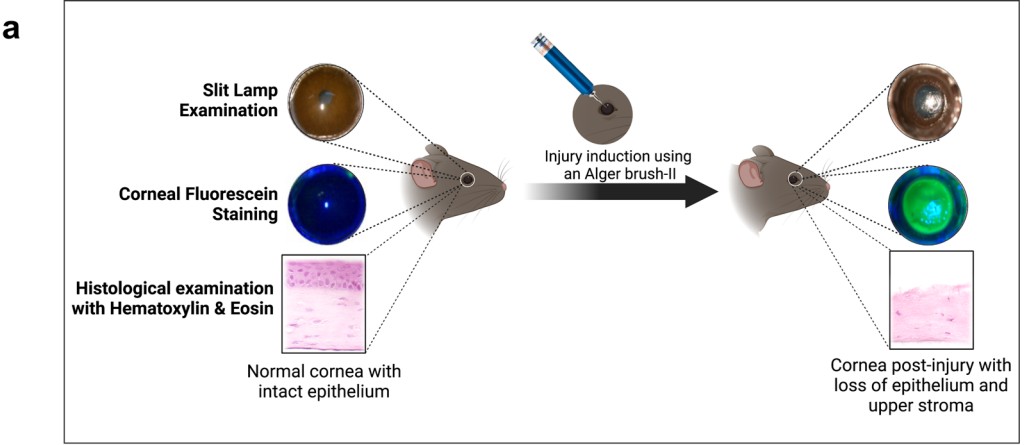

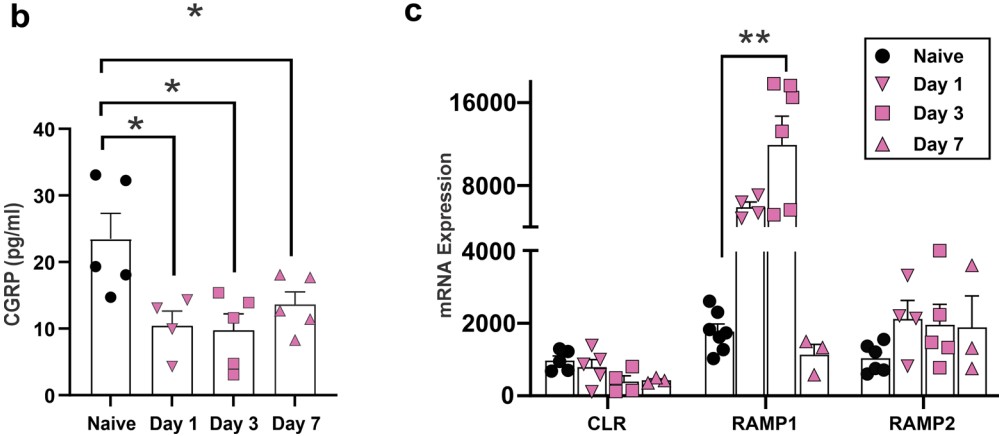

**Fig. 1 | Mechanical injury causes a decrease in CGRP levels in the cornea.**
**a** Schematic figure showing the method of mechanical injury. The epithelium and superficial stroma were removed using Algerbrush-II. **b** Assessment of CGRP protein levels using ELISA showed significantly reduced expression following corneal injury on days 1, 3, and 7. **c** The gene expression levels of the CGRP receptors, calcitonin receptor-like receptor (CLR), receptor activity-modifying protein (RAMP) 1, and RAMP2. RT-PCR showed upregulation of the RAMP1 on day 3 post-injury and returned to normal levels by day 7. ($n = 3–5$ per group). The data were presented as mean ± standard error of mean (SEM) and comparison is determined by a one-way ANOVA test with pairwise comparison. *$p < 0.05$, **$p < 0.01$. (Legend: Circle = naive, Inverted Triangle = Day 1, Square = Day 3, Upright Triangle = Day 7). Figure 1a was created with BioRender.com.

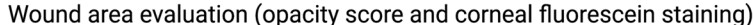

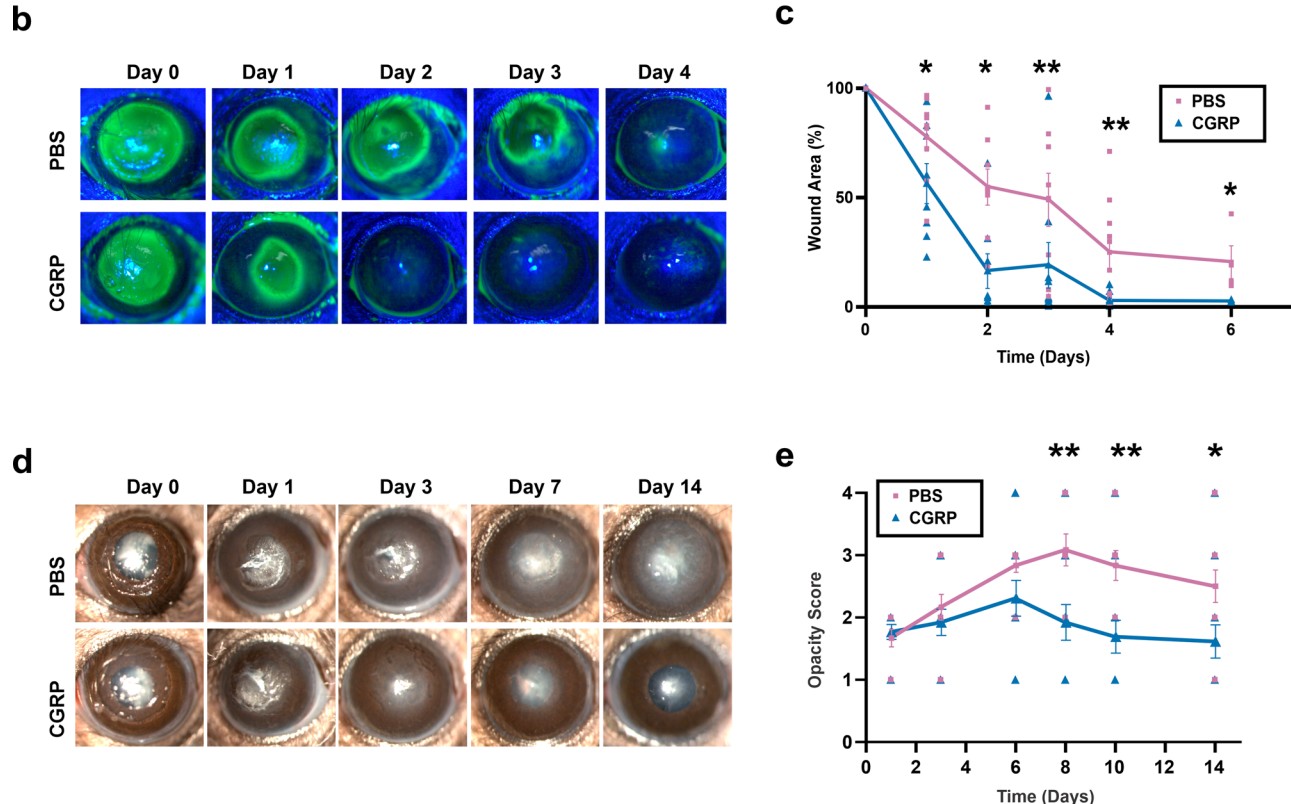

**Fig. 2 | Topically applied CGRP accelerates corneal epithelial wound closure and reduces corneal opacity. a** Schematic figure showing the timeline of treatment and clinical examination following injury. **b** Corneal fluorescein staining was performed to compare the size of epithelial defects in CGRP- and PBS-treated mice at different time points post-injury. **c** The assessment of the wound area shows that CGRP treatment resulted in a significantly smaller wound area compared to PBS-treated controls across all time points from 24 h to 6 days ($n = 9$ per group). **d** Representative slit lamp photographs of PBS and CGRP-treated eyes up to 14 days after injury. The corneas of the PBS-treated controls showed progressive stromal opacification, whereas CGRP treatment showed significantly lower corneal opacity. **e** The scoring of corneal opacity was performed in a blinded fashion and showed a significantly lower score in CGRP-treated mice ($n = 12$ per group). The data were represented as mean ± SEM. The statistical significance was determined by unpaired $t$-test, *$p < 0.05$, **$p < 0.01$. (Legend: Pink Square = PBS, Blue Triangle = CGRP). Figure 2a was created with BioRender.com.

$p = 0.006$), day 10 ($2.8 \pm 0.2$ vs $1.7 \pm 0.3$, $p = 0.004$) and day 14 ($2.5 \pm 0.3$ vs $1.6 \pm 0.2$, $p = 0.03$) (Fig. 2d, e).

## CGRP decreases corneal thickness, scar formation, and endothelial cell loss after injury

In addition to slit lamp photography, the animals underwent live imaging with optical coherence tomography (OCT) and confocal microscopy to visualize corneal microstructure in vivo. Shown in Fig. 3a, anterior segment-OCT demonstrates the normal mouse eye anatomy. In the control eyes post-injury, we observed hyperreflectivity of the corneal stroma, consistent with

increased corneal opacity on slit lamp exam; in addition, the central corneal thickness (CCT) increased from $90.5 \pm 1.5$ at baseline to $159.6 \pm 16.8$ μm on day 14 (Fig. 3a, b). The CGRP-treated corneas showed lower CCT compared to the PBS-treated injured mice on days 7 ($100.5 \pm 8.5$ vs $166.8 \pm 11.7$ μm, $p < 0.0001$) and 14 ($97.6 \pm 6.7$ μm vs $159.6 \pm 16.8$ μm, $p = 0.0008$) (Fig. 3b).

In vivo confocal microscopy (IVCM) reveals the cellular structure of the cornea; and naïve animals have regular and densely packed polygonal epithelial and endothelial cells, in addition to uniform reflectivity in the stroma (Fig. 3c). The control corneas demonstrated reduced uniformity and density of corneal epithelial and stromal cells, as well as hyperreflectivity and

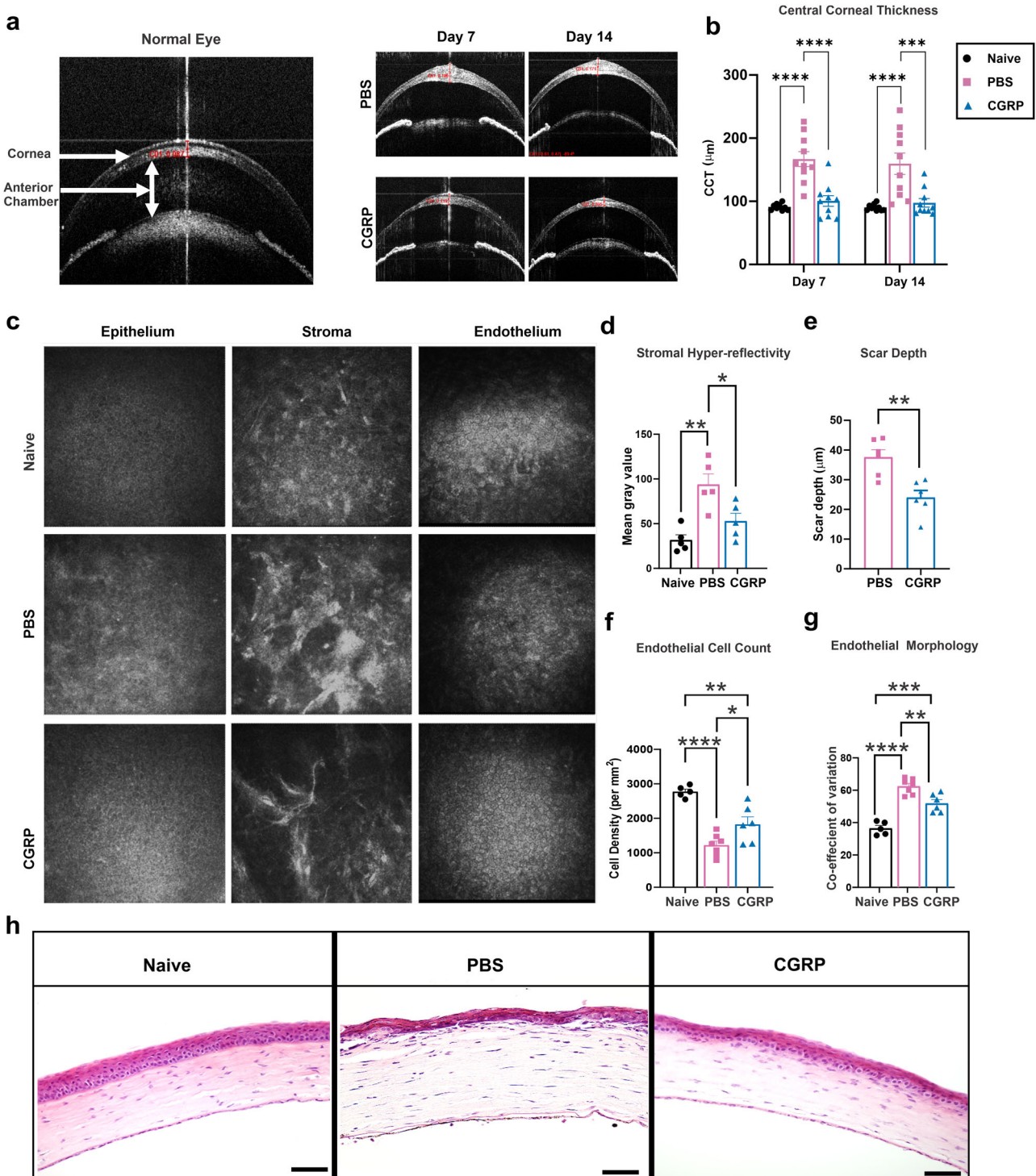

**Fig. 3 | CGRP decreases corneal thickness, scar formation, and endothelial cell loss after injury. a** Representative AS-OCT images showed a significant increase in central corneal thickness (CCT) and stromal hyperreflectivity in PBS-treated mice, whereas they were comparable to naïve mice in the treated group on days 7 and 14 post-injury. **b** CGRP treatment resulted in significantly lower CCT compared to PBS-treated controls on days 7 and 14 (*n* = 10 per group). **c** Representative IVCM images of the corneal epithelium, stroma, and endothelium. The analysis of IVCM images showed that CGRP treatment resulted in decreased stromal hyperreflectivity (**d**), scar depth (**e**), endothelial cell loss (**f**), and endothelial coefficient of variation (**g**) compared to PBS-treated mice at day 14 post-injury. **h** The histological analysis by hematoxylin and eosin staining showed reduced corneal thickness and inflammatory cell infiltration following CGRP compared to the PBS treatment (scale bar = 200 μm). **d** *n* = 5, **e**–**g** *n* = 6 per group. The data were represented as mean ± SEM, and the statistical significance was determined by one-way ANOVA (**b, d, f, g**) and unpaired *t*-test (**e**), *$p < 0.05$, **$p < 0.01$, ***$P < 0.001$, ****$P < 0.0001$. (Legend: Black Circle = Naïve, Pink Square = PBS, Blue Triangle = CGRP).

dense scars in the stroma, on day 14 post-injury. The CGRP-treated corneas showed comparable micro-anatomy to the uninjured corneas, except mild scar formation in the stroma (Fig. 3c). Objective assessment of the stromal hyperreflectivity (Fig. 3d) confirmed these observations and the depth of the scar in the CGRP group is less deep compared with the controls (Fig. 3e). The corneal endothelial cell density and morphology (demonstrated by coefficient of variation of cell size and the percentage of normal hexagonal cells) were better maintained in the CGRP-treated eyes, compared to the controls (Fig. 3f, g and Supplementary Fig. 1a).

Similarly, histological assessment with H&E staining demonstrated that mechanical injury led to overall thickening of the cornea, thinner epithelium, separation between the epithelium and anterior stroma, inflammatory cell infiltration, disorganized stroma, and attenuated endothelium in the control animals. On the contrary, the corneal structure and integrity were much better maintained in the CGRP-treated mice (Fig. 3h).

To determine the short-term effect of CGRP on injury repair, we applied CGRP for 5 days only until the epithelial defect closed. We found that the CGRP-treated group exhibited a significantly lower opacity score compared to the PBS-treated group. However, this short course of CGRP treatment failed to improve corneal edema, as compared to the 14-day treatment (Supplementary Fig. 2). This suggests that the effect of CGRP on epithelial closure and corneal opacity can be achieved with a short treatment course, whereas its therapeutic effect on suppression of inflammation and thus protection of corneal endothelial cells requires continuous and longer treatment course.

To delineate the mechanisms underlying the regenerative function of CGRP at cellular and molecular levels, we next sought to determine its effects on corneal epithelial, stromal, endothelial, and inflammatory cells in vitro and in vivo.

## CGRP promotes corneal epithelial cell regeneration in vitro and in vivo

After confirming the expression of the CGRP receptor components CLR, RAMP1, and RAMP2 in the human corneal epithelial cells (hCEC) (Supplementary Fig. 3a), we cultured the hCEC and treated them with increasing concentrations of CGRP (10 to 1000 nM). We observed a dose-dependent effect of CGRP on hCEC proliferation with increasing frequency of Ki67-positive cells in the CGRP-treated groups (Fig. 4a, b). Next, we assessed cell migration using a scratch assay into a confluent monolayer of hCEC and showed a dose-dependent increase in wound closure (Fig. 4c, d). One key function of CEC is to secrete laminin, a main component of the EBM[35]. We observed a significantly higher level of laminin gene expression in hCEC with CGRP treatment (Fig. 4e). ERK1/2 have been shown as critical cell signaling molecules that regulate CEC function. Using immunostaining and western blotting, we next demonstrated that CGRP rapidly (as early as 1 h) activates (phosphorylates) ERK1/2 in hCEC. (Fig. 4f, g). These effects of CGRP on cultured hCEC were confirmed in vivo (Fig. 4h, i). We collected mouse corneas on day 4 post-injury and observed high levels of Ki67 and laminin staining in the CGRP-treated corneas, indicating that CGRP treatment promotes corneal epithelial cell proliferation and basement membrane formation post-injury in vivo.

## CGRP suppresses TGF-β1 signaling and corneal stromal fibroblast activation in vitro and in vivo

Fibroblasts are the primary stromal cells involved in scar formation following corneal injury and TGF-β1 is a key promoter of fibroblast activation[7,36,37]. As we observed reduced corneal opacity clinically and stromal hyperreflectivity via IVCM in CGRP-treated animals, we next sought to determine its effect using the corneal fibroblast cell line MK/T1. After confirming its expression for CGRP receptors (Supplementary Fig. 3b), we induced fibroblast activation with TGF-β1 and observed that CGRP significantly suppressed the expression of alpha-smooth muscle actin (α-SMA) in MK/T1 cells using RT-PCR (Fig. 5a), western blotting (Fig. 5b and Supplementary Fig. 4a), and immunostaining (Fig. 5c).

The in vitro suppressive effect of CGRP on fibroblast activation was confirmed in vivo. Mouse corneas were collected on day 5 post-injury. Compared to the uninjured naïve corneas, PBS-treated injured corneas expressed much higher levels of TGF-β1 and CGRP treatment led to a decrease in its level, but it was still higher than the naïve corneas in RT-PCR and western blot (Fig. 5d, e and Supplementary Fig. 4b). Interestingly, immunohistochemical analysis at day 14 showed minimal TGF-β1 staining in the CGRP-treated corneas (Fig. 5f), comparable to the injured controls. Similarly, we observed that CGRP treatment reduced the injury-promoted α-SMA expression in the corneal stroma in vivo using RT-PCR and western blot at day 5 and immunostaining at day 14 (Fig. 5f–h and Supplementary Fig. 4c)

## CGRP preserves corneal endothelial cell density and function in vivo

Previous studies have shown that ocular trauma can induce corneal endothelial cells (CEnC) swelling and loss[38,39] The loss or dysfunction of these cells leads to corneal edema and a decrease in vision[40]. To study the effect of the mechanical injury and CGRP treatment on the CEnC, we collected mouse corneas on day 14 post-injury and stained the corneal endothelium with zonula occludens (ZO-1), a tight junction protein critical for CEnC function and conveniently outlining cell border, and Na$^+$/K$^+$-ATPase (Fig. 6a). Similar to the IVCM findings, naïve uninjured cornea endothelium displays hexagonality and uniformity in morphology with Na$^+$/K$^+$-ATPase staining of cell membrane. In the injured PBS-treated cornea, the continuous ZO-1 staining pattern was disrupted, the cells appeared large and variable in size, and the Na$^+$/K$^+$-ATPase staining pattern was irregular. The CGRP-treated eyes, however, showed ZO-1 and Na$^+$/K$^+$-ATPase staining patterns similar to the uninjured eyes (Fig. 6a–c and Supplementary Fig. 1b). Additionally, we found increased expression of the α1 and α3 subunits of Na$^+$/K$^+$-ATPase by CGRP treatment, compared to the control (Fig. 6d).

Having observed the positive effects of CGRP on CEnC, we sought to investigate whether topically applied CGRP could penetrate the corneal stroma and Descemet's membrane to reach CEnC. We quantified CGRP concentration in the aqueous humor 30 min after CGRP administration. However, no change in CGRP concentration was observed in naïve mice or on day 1 or day 7 post-injury following CGRP administration (Supplementary Fig. 5). This implies that the effect of CGRP on CEnC observed in vivo is likely via modulating the microenvironment (such as reducing tissue inflammation) rather than exerting a direct effect on CEnC.

## CGRP dampens tissue inflammation after injury

As CGRP has been shown to modulate the innate immune response[41–43], we sought to determine its effect on corneal inflammation in vivo. On day 1 post-injury, there was a notable increase in the infiltration of CD45$^+$ cells in the cornea in the injured PBS-treated controls (9.7 ± 1.0% of all corneal cells) compared to uninjured naïve corneas (0.07 ± 0.04%). This effect persisted up to day 3 post-injury (2.1 ± 0.3% of all corneal cells). Treatment with topical CGRP resulted in a reduction of CD45$^+$ cell infiltration, decreasing to 5.7 ± 0.4% on day 1 and further to 1.2 ± 0.1% on day 3 post-injury (Fig. 7a, b). However, it is important to note that despite an ~40% reduction in immune cell infiltration after injury, CGRP-treated corneas still had significantly higher CD45$^+$ cell infiltration than the naïve eyes. This result suggests that CGRP mitigates, rather than completely inhibits, the innate immune response, which is crucial for proper wound healing.

Furthermore, the increased expression of pro-inflammatory chemokine (C-X-C motif) ligand 1 (CXCL1) after injury was significantly reduced in the CGRP-treated corneas (Fig. 7c). Cytokines TNF and interleukin-1β (IL-1 β), as well as pro-inflammatory mediator matrix metalloproteinase 9 (MMP-9) in the cornea, were upregulated in the PBS-treated corneas, and CGRP treatment significantly reduced their levels on day 3 post-injury (Fig. 7d).

The analysis of the infiltrating CD45$^+$ cells in the cornea post-injury revealed that most of these cells were either neutrophils (CD11b$^+$Ly6G$^+$) or

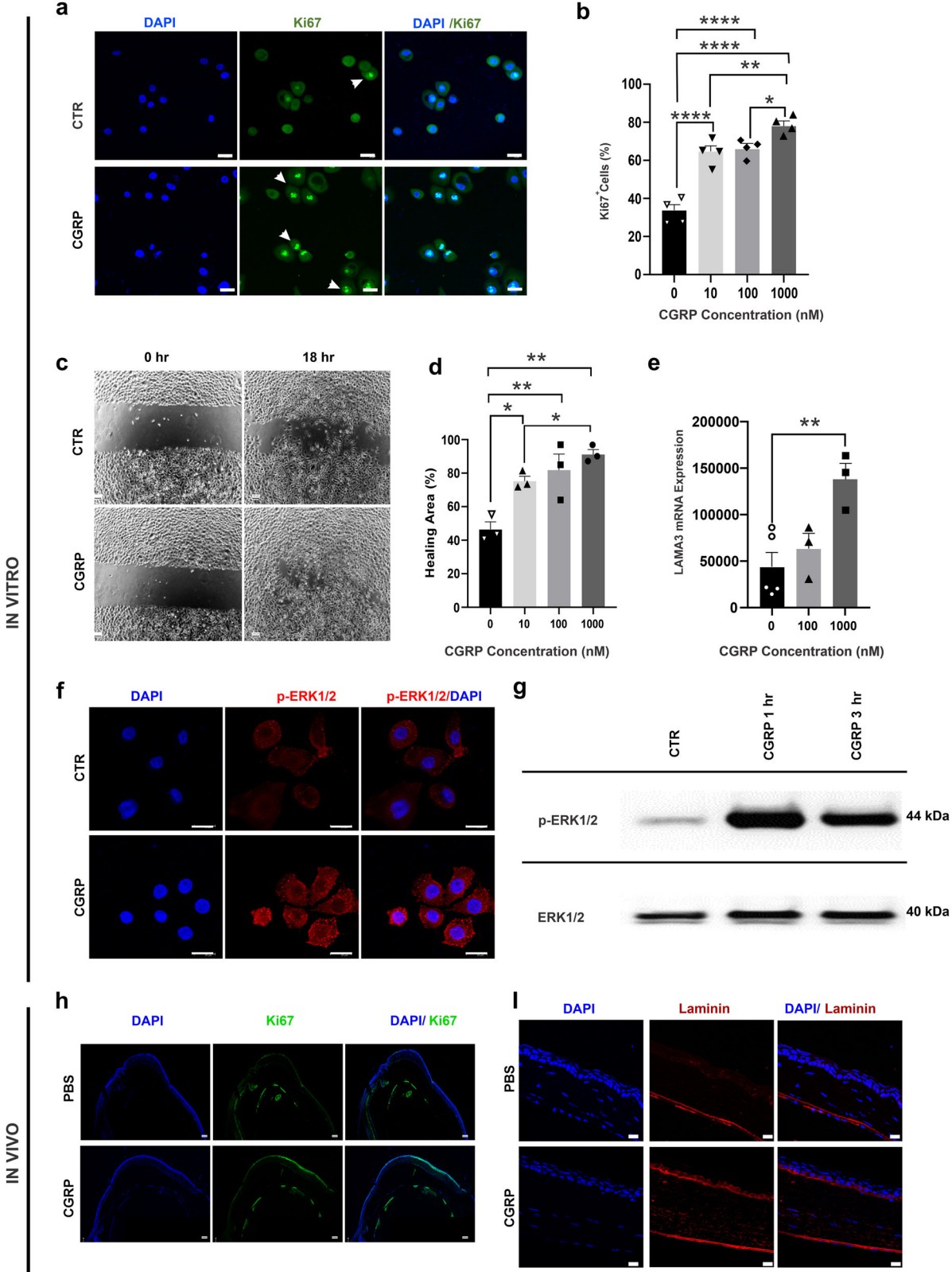

**Fig. 4 | CGRP promotes corneal epithelial cell (CEC) regeneration in vitro and in vivo. a** Human CEC were cultured and CGRP (1 μM for 24 h) led to an increased frequency of proliferating Ki67+ cells (arrowheads, green) (scale bar = 20 μm). **b** CGRP resulted in an increase in the Ki67+ cells in a dose-dependent manner (*n* = 4 per group). **c** CEC were cultured to a monolayer and a linear scratch was created. **d** CGRP promoted CEC migration in a dose-dependent manner (*n* = 3 per group). **e** RT-PCR data showed significantly increased *laminin 332* expression in CEC by CGRP (*n* = 3 per group). **f** CGRP (1 μM for 1 h) increased the phosphorylation of ERK (p-ERK antibody, red, (scale = 20 μm). **g** Western blot analysis

confirmed the increased p-ERK levels with CGRP treatment. **h** Mouse corneas obtained (on day 4) from the CGRP-treated mice showed more Ki67 (green) staining in the epithelium compared to PBS-treated controls, (scale bar = 100 μm). **i** A higher level of laminin immunostaining (red) was also observed in the cornea derived from CGRP-treated mice compared to the PBS-treated controls (scale bar = 20 μm). The data were presented as mean ± SEM, and the statistical significance was determined by one-way ANOVA with pairwise comparison, *$p < 0.05$, **$p < 0.01$, ****$P < 0.0001$.

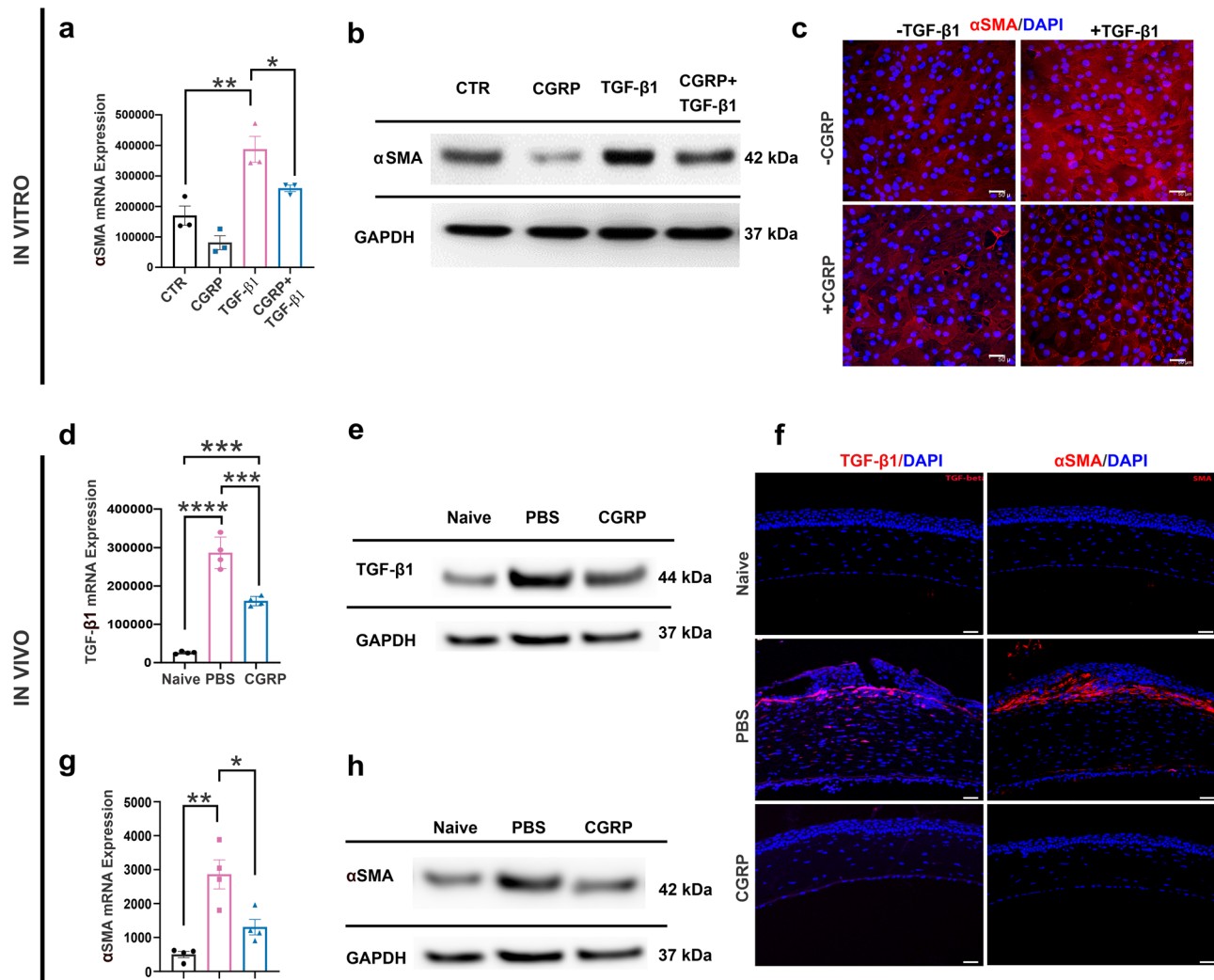

**Fig. 5 | CGRP suppresses TGF-β1 signaling and corneal stromal fibroblast activation in vitro and in vivo.** Murine corneal fibroblasts were cultured in a medium supplemented with 10 ng/ml TGF-β1 for 24 h, in the presence or absence of 1 µM CGRP. CGRP significantly decreased the TGF-β1-mediated expression of α-SMA assessed via RT-PCR (**a**), western blot (**b**), and immunostaining (**c**) in vitro (*n* = 3 per group). Corneas derived from CGRP-treated mice showed significantly lower expression of TGF-β1 compared to PBS-treated controls in vivo in RT-PCR

(**d**, *n* = 4 per group), western blot (**e**), and immunostaining (**f**, scale = 50 µm). Similarly, α-SMA levels in RT-PCR (**g**, *n* = 4 per group), western blot (**h**), and immunostaining (**f**) were significantly elevated post-injury and decreased by CGRP treatment in vivo. The data were presented as mean ± SEM, and the statistical significance was determined by one-way ANOVA with pairwise comparison. *$p < 0.05$, **$p < 0.01$, ***$P < 0.001$, ****$P < 0.0001$. (Legend in **d**, **g**: Black Circle = Naïve, Pink Square = PBS, Blue Triangle = CGRP).

macrophages (CD11b+Ly6G−). Topical CGRP treatment significantly reduced the frequency of the infiltrating neutrophils to 0.6 ± 0.1% (966.3 ± 246.4 cells/cornea) compared to PBS-treated controls 1.6 ± 0.1% (3081 ± 505.2 cells/cornea) (Fig. 7e, f). Interestingly, the frequency of macrophages was comparable in the two groups with 0.38 ± 0.02% (301.5 ± 122.5 cells/cornea) in the PBS-treated group and 0.30 ± 0.1% (261 ± 175 cells/cornea) in the CGRP treatment group (Fig. 7e, f). We then assessed the expression levels of macrophage maturation markers, major histocompatibility complex-II (MHC-II), C-C chemokine receptor type 2 (CCR2), and inducible nitric oxide synthase (iNOS), and found that they were significantly reduced by CGRP treatment compared to the PBS-treated controls (Fig. 7g)

## Discussion

In this research report, we evaluated the effect of CGRP on corneal wound healing following mechanical injury (Fig. 8). Corneal injury leads to nerve damage and depletion of neuropeptides including CGRP. Topical application of CGRP as an eye drop accelerates corneal epithelial closure, preserves corneal transparency, and prevents scar formation and edema.

Mechanistically, CGRP promotes corneal epithelial cell migration, proliferation, and the secretion of the basement membrane; it reduces TGF-β1-mediated stromal fibroblast activation and tissue fibrosis; CGRP preserves corneal endothelial density and function; and lastly, it reduces neutrophil infiltration, macrophage maturation, and the production of inflammatory cytokines. Taken together, our results show that corneal nerve-derived CGRP plays a regenerative, antifibrotic, and anti-inflammatory role in corneal wound healing and that corneal innervation is a key regulator of ocular surface homeostasis and injury repair.

The cornea is a densely innervated tissue and the concentrations of neuropeptides secreted by these nerves change in response to various pathologies[44,45]. The 37-amino acid long neuropeptide CGRP is a member of the calcitonin peptide superfamily, produced by the alternate splicing of the calcitonin gene, and secreted in the central and peripheral nervous systems including corneal nerves[46,47]. Previous studies have reported that CGRP levels changed after ocular surface trauma and inflammation[32]. Lambaise et al. observed that CGRP concentration significantly decreased in the tear film of dry eye disease patients and that its level correlated with the disease severity[48]. Following photorefractive keratectomy, CGRP concentration in

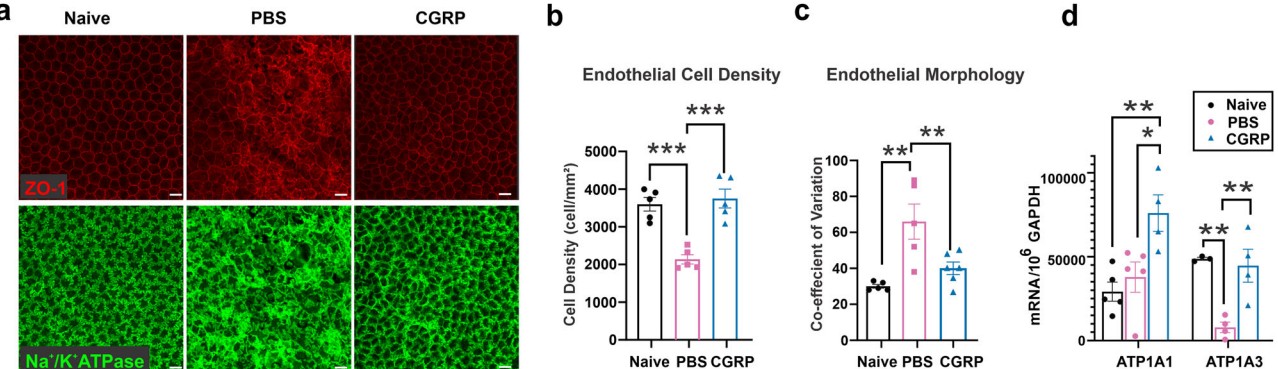

**Fig. 6 | CGRP preserves corneal endothelial cell density and function in vivo.**
**a** Mouse corneas were collected on Day 14 post-injury and CGRP treatment led to preserved zonula occludens-1 (ZO-1) and Na$^+$/K$^+$ ATPase staining, compared to the PBS-treated controls (scale = 20 μm). Analysis of immunohistochemical images showed higher endothelial cell density (**b**) and lower coefficient of variation (**c**) in vivo (n = 6 per group). **d** RT-PCR evaluation of mouse corneas showed higher gene expression of α1 and α3 isoforms of Na$^+$/K$^+$ ATPase in CGRP-treated mice compared to controls (n = 3 per group). The data were presented as mean ± SEM, and the statistical significance was determined by one-way ANOVA with pairwise comparison (**b–d**), *$p < 0.05$, **$p < 0.01$, ***$P < 0.001$. (Legend: Black Circle = Naïve, Pink Square = PBS, Blue Triangle = CGRP).

the tear film fluctuated with an initial increase during the healing phase (on day 2) followed by a decline on day 7[32,33].

To date, there is no report on the level of CGRP in the cornea after injury or surgery. We found that removal of the epithelial and upper stromal layer of the cornea leads to a marked decrease in CGRP concentration in the cornea up to 7 days post-injury. Since CGRP is exclusively expressed by the nerves in the cornea[18], we attribute its reduced level to the loss of CGRP-expressing corneal nerves. Notably, we also observed an upregulation of CGRP receptor component RAMP1 in the cornea during the same period (Fig. 1c). We speculate that this is due to either the infiltration of RAMP1-expressing immune cells into the cornea or compensatory upregulation[17]. Interestingly, we noted an increase in CGRP concentration in the aqueous humor on day 1 after injury, followed by a decline on day 7 (Supplementary Fig. 5). The source of CGRP in the aqueous humor and this increase could be potentially linked to sensory nerve stimulation[49] in the ciliary body[50–52] and warrants further investigation.

Several studies have highlighted the critical role of CGRP in wound healing and its absence (in CGRP knockout mice or antibody-mediated blockade) results in delayed wound closure[53] and accentuated inflammation[54]. We observed that CGRP promotes corneal epithelial proliferation and migration in vitro and in vivo. Our observations are consistent with the report by Mikulec et al. that CGRP increased epithelial healing by 25% in a whole mount corneal preparation derived from the rabbits in vitro[34]. Similar regenerative effects (proliferation and migration) of CGRP were also observed in bronchial epithelial cells in a dose-dependent manner via MAP kinase (MAPK) pathway[23]. The phosphorylation of MAPKs such as ERK1/2 is essential for cell proliferation and migration[55,56]. Indeed, we found that CGRP treatment leads to rapid and robust phosphorylation of ERK1/2 in the corneal epithelial cells.

Wound healing is attributed to the regenerative potential of epithelial cells; in addition, these cells secrete components of the epithelial basement membrane (EBM), which plays an important role in maintaining epithelial integrity and more importantly limits the impact of trauma on the superficial cornea[35]. EBM is primarily composed of type IV collagen, proteoglycan, and glycoproteins (laminin, fibronectin, nidogen, entactin)[35,57]. The apoptosis of epithelial cells following trauma leads to the release of growth factors and pro-inflammatory mediators including TGF-β1, TNF, and IL-1[7]. An intact EBM prevents these pro-inflammatory mediators from leaking into the underlying stroma[7]. On the contrary, the breakdown of the EBM disrupts the barrier between the apoptotic cells and the stroma, thus exposing the underlying stromal keratocytes to differentiate into myofibroblast, which in turn triggers the activation of more keratocytes in the vicinity[8,58]. In our in vitro experiments, we observed that CGRP induces the epithelial expression of laminins, which are the glycoproteins expressed in

the basal lamina and play an important role in EBM integrity (Fig. 4). Similarly in vivo CGRP treatment promotes laminin expression, thereby restoring the EBM integrity and interrupting the TGF-β1 induced differentiation of fibroblasts. Consistent with this, we observed much lower TGF-β1 levels in the CGRP-treated corneas in vivo. TGF-β1-mediated production of α-SMA, a marker of activated myofibroblasts, is also reduced by CGRP treatment in vitro and in vivo. These suppressive effects of CGRP on fibroblast activation are clinically observed as well, as the mice treated with the CGRP show less corneal opacification and scar formation. A similar antifibrotic effect of CGRP has been previously reported in models of cardiac and pulmonary fibrosis[59,60].

The inflammatory milieu in the cornea post-trauma is another critical component impacting the outcomes of wound healing[8]. It is known that injured epithelial cells and the apoptotic keratocytes secrete factors to activate the innate immune system and promote the release of pro-inflammatory mediators, which induce the activation of monocytes to macrophages and the recruitment of neutrophils to the site of injury. These leukocytes further aggravate inflammation by releasing cytokines and chemokines and inducing cell death/apoptosis, creating a vicious cycle[61]. Although the innate immune response is primarily tasked with clearing cellular debris and infectious organisms, persistent tissue inflammation may lead to tissue damage[62]. We observed that CGRP dampens the inflammatory reaction by reducing leukocyte infiltration and reducing the expression of the pro-inflammatory mediators IL-1β, TNF, and MMP-9.

The expression of CXCL1, which plays a predominant role in neutrophils recruitment, is decreased by CGRP treatment in vivo, leading to reduced neutrophils infiltration (Fig. 7). These results align with previous reports on the anti-inflammatory effect of CGRP[63,64]. Interestingly, we observed a comparable frequency of macrophages in corneas derived from CGRP- and PBS-treated mice. However, CGRP treatment results in decreased iNOS and CCR2 expression in macrophages. iNOS and CCR2 are markers for pro-inflammatory macrophage sub-population that are typically seen in corneal tissue following injury[65]. Additionally, CGRP treatment leads to lower expression of MHC class II, indicating decreased macrophage maturation. Our findings on the effects of CGRP on macrophages are consistent with a report by Duan et al. that CGRP promoted polarization of macrophages from M1 to M2[43]. Further study is needed to characterize the macrophage sub-population in greater detail.

The inflammatory milieu post-injury not only induces stromal fibrosis in the cornea but also leads to corneal endothelial cell (CEnC) loss and dysfunction[10]. Previous clinical studies have reported such impact of the trauma on the CEnC leading to their swelling and disruption with fibrin and leukocyte accumulation[39]. Moreover, measurable decreases in CEnC density, up to 20%, were noted in patients with a history of blunt ocular

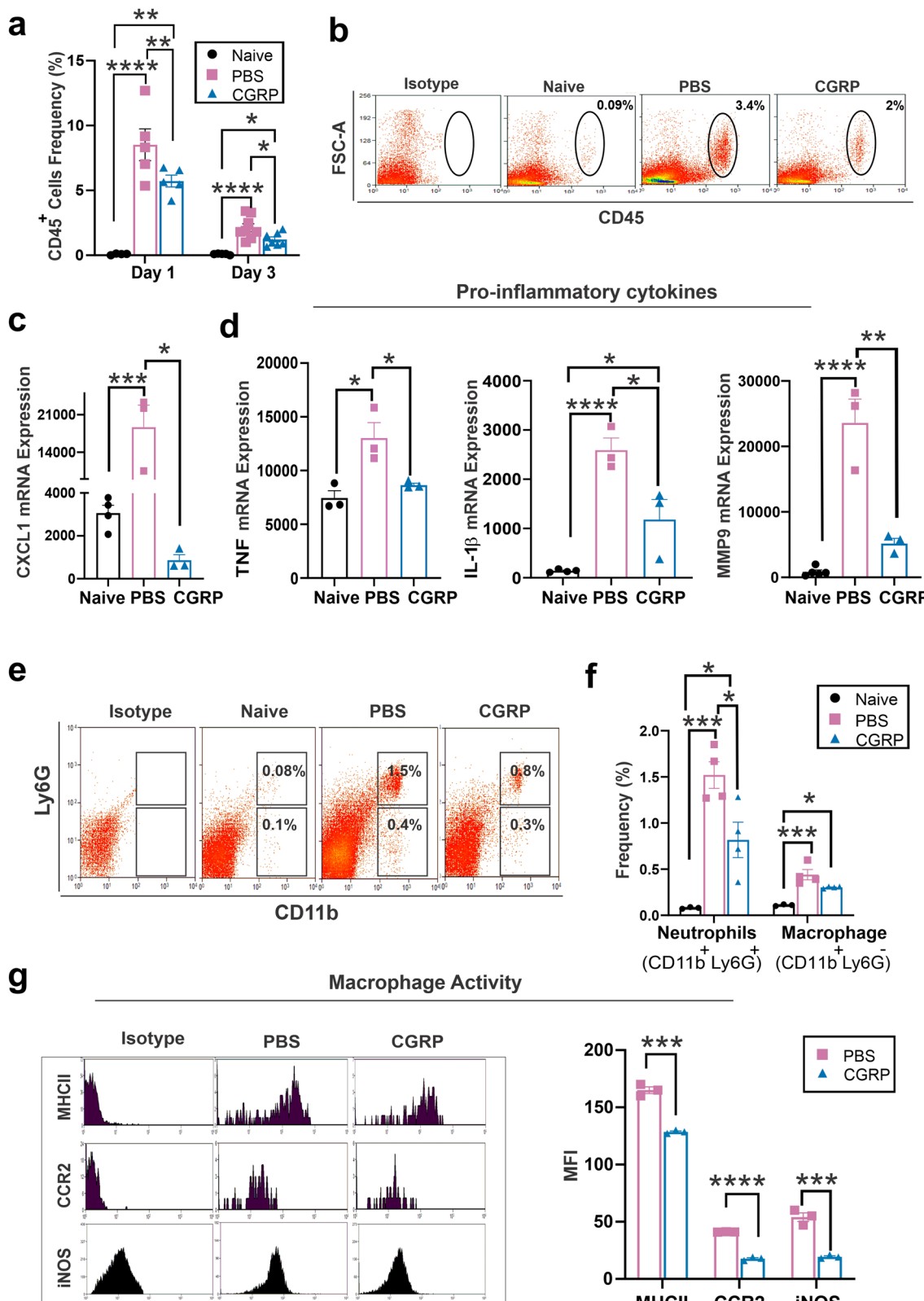

trauma[38,66–68]. Similarly, we observe a significant decrease in endothelial cell density, altered cellular morphology, and reduced expression of pump protein $Na^+/K^+$-ATPase after injury, resulting in persistent corneal edema (increased central corneal thickness) in vivo. CGRP topical application maintains CEnC density, morphology, and corneal deturgescence in vivo. Interestingly, topical application of CGRP does not change its concentration in the aqueous humor, whether in naïve or injured eyes. Therefore, the observed CEnC cytoprotection by CGRP is likely indirect and via its ability to dampen inflammatory response, which has deleterious effects on the CEnC[68–71].

It is worth noting that CGRP administration (intracerebroventricular, intrathecal, intravenous, intraperitoneal) leads to migraine-like phenotypes

**Fig. 7 | CGRP dampens tissue inflammation after injury in vivo. a** Lower frequencies of CD45$^+$ cells in corneas derived from CGRP-treated mice compared to controls on day 1 and day 3 post-injury, ($n = 5$ per group). **b** Representative flow cytometry plots show that CGRP treatment suppressed the infiltration of CD45$^+$ into the cornea on day 3 post-injury compared to PBS-treated controls. The RT-PCR analysis showed that the CGRP treatment resulted in significantly lower expression of CXCL1 (**c**), IL-1β, TNF, and MMP-9 (**d**) on day 3 post-injury ($n = 3$ per group). **e** Representative flow cytometry plot shows the frequencies of neutrophils (CD11b$^+$Ly6G$^+$) and macrophage (CD11b$^+$Ly6G$^-$) in the mice cornea. **f** The

frequencies of neutrophils were significantly lower in corneas derived from CGRP-treated mice compared to controls, and the frequencies of macrophages were comparable in the two groups ($n = 3$ per naive group, $n = 4$ per PBS and CGRP group). **g** CGRP treatment resulted in significant suppression of MCH-II, CCR2, and iNOS expression in CGRP-treated mice compared to the controls ($n = 3$ per group). The data were presented as mean ± SEM, and the statistical significance was determined by one-way ANOVA with pairwise comparison (**a–f**) and unpaired $t$-test (**g**), *$p < 0.05$, **$p < 0.01$, ***$P < 0.00$, ****$P < 0.0001$. (Legend: Black Circle = Naïve, Pink Square = PBS, Blue Triangle = CGRP).

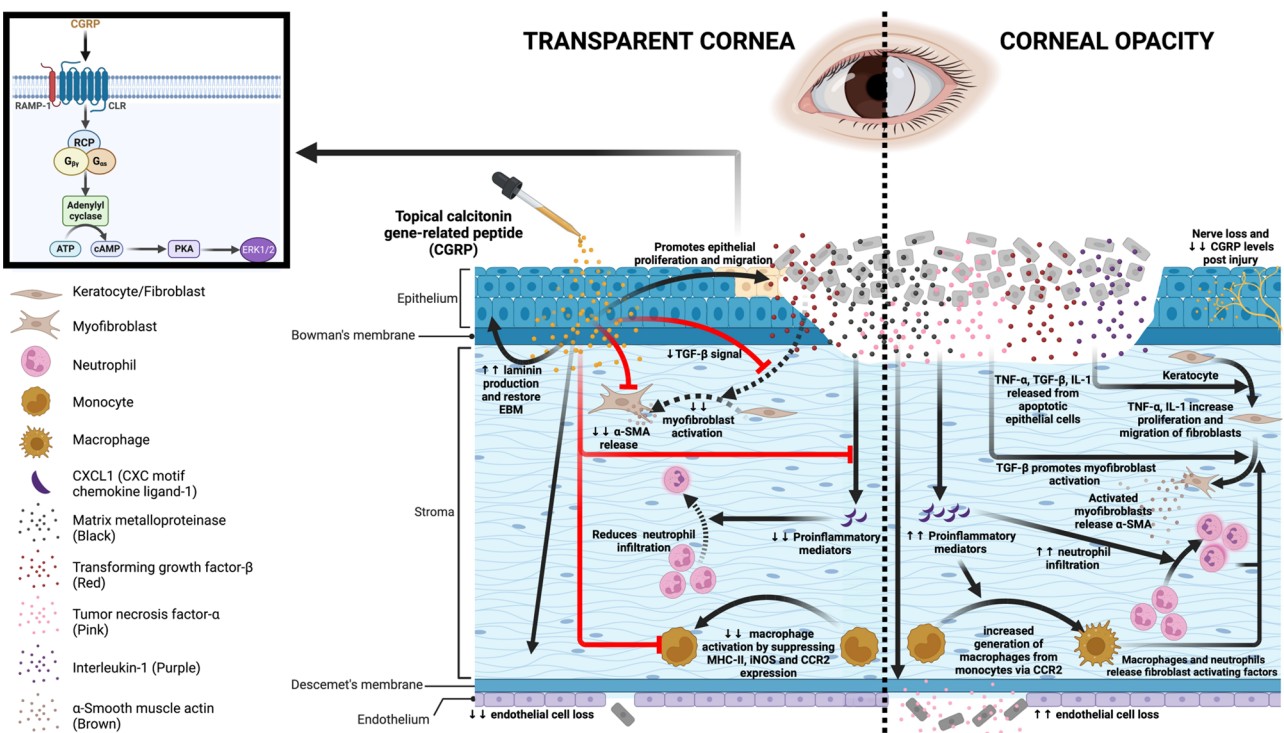

**Fig. 8 | Schematic showing the effects of CGRP on corneal wound healing after mechanical injury.** Injury leads to nerve damage and a decrease in CGRP level in the cornea. Topical application of CGRP promotes corneal epithelial cell regeneration and restores the epithelial basement membrane, thus reducing the release of pro-inflammatory and pro-fibrotic mediators including TNF-α, TGF-β, IL-1, and CXCL1 into the stroma. This leads to reduced keratocyte activation and stromal fibrosis. In addition, CGRP reduces neutrophil infiltration, macrophage maturation, and the production of inflammatory cytokines. It reduces corneal endothelial cell loss and maintains its pump function. Clinically, topical application of CGRP as an eye drop accelerates epithelial closure, preserves transparency, and prevents scar formation and edema after corneal injury. The figure was created with BioRender.com.

through central and peripheral sensitization[72]. However, there has not been a report on the induction or development of migraine after ophthalmic application of CGRP to our knowledge. In our current study, we found increased ocular pain response after corneal injury up to 14 days post-injury (Supplementary Fig. 6), consistent with prior reports[73,74]. Interestingly, topical application of CGRP did not heighten pain response at any time point observed. Rather, it decreased the pain response at Day 3 post-injury, likely due to accelerated epithelial healing. Nevertheless, further elucidation is needed to understand the impact of the ophthalmic application of CGRP on migraine headache.

In conclusion, corneal wound healing is a complex process that requires an integrated response to restore its transparency. We found that topical treatment with CGRP promotes tissue repair and limits injury-induced corneal opacity by enhancing epithelial cell proliferation and migration, restoring epithelial basement membrane integrity, suppressing TGF-β1-mediated tissue fibrosis, decreasing neutrophil infiltration and macrophage activation, inhibiting expression of pro-inflammatory mediators, and preserves corneal endothelial cell density and function (Fig.8). Collectively this proof-of-concept murine study demonstrates the efficacy of CGRP in treating corneal mechanical injury

and provides evidence on the essential role of sensory nerves in tissue would healing.

## Methods
### Animal

Six- to eight-week-old male and female C57BL/6 mice were purchased from Charles River Laboratories (Wilmington, MA, USA). All the animal experiments were approved by the Schepens Eye Research Institute Animal Care and Use Committee and were performed in strict adherence of the Association for Research in Vision and Ophthalmology Statement for the Use of Animals in Ophthalmic and Vision Research. The animals were provided post-operative pain management and care as per requirement. We have complied with all relevant ethical regulations for animal use.

### Corneal mechanical injury model
We used a well-established murine model of corneal mechanical injury as described previously[75]. Briefly, the central area of the cornea was marked with a 2-mm trephine. The epithelial layer and the superficial stroma were removed using a hand-held Algerbrush-II (Alger Inc., Lago Vista, TX). The injured corneas were washed with sterile phosphate-buffered saline (PBS) to

remove any tissue debris. Post-injury induction, mice were randomly divided into two groups; the first group was topically administrated CGRP (5 µl of a 50 µM stock solution diluted in 1X PBS) three times daily, while the second group received topically administrated PBS in the same volume served as controls. We assessed corneal opacity and thickness by slit lamp biomicroscope (Topcon, Tokyo, Japan) and anterior segment-optical coherence tomography (AS-OCT, Envisu R2200 Spectral Domain Ophthalmic Imaging System with 12 mm telecentric, Bioptigen Inc., NC, USA), respectively. The micro-anatomy of the cornea was visualized using in vivo confocal microscopy (IVCM, Heidelberg Retina Tomograph HRT III with Rostock Corneal Module, Heidelberg GmbH, Germany). The images of the endothelium were analyzed for cell density (CD, cells/mm$^2$), hexagonality percentage (Hex), and the coefficient of variation (CV, amount of variation in cell size) using CellChekD+ software (Konan Medical).

### Topical CGRP treatment

We prepared topical eye drops from CGRP lyophilized powder (Bachem, Bubendorf, Switzerland, Cat. No. 4025897). The powder was reconstituted in sterile PBS at the final concentration of 50 µM. The eye drop solution was aliquoted and stored at −20 °C for long-term use and thawed before the experiment. For in vivo experiments, the animals in the CGRP treatment group were instilled with 5 µl of eye drops, three times a day for 14 days.

### Corneal fluorescein staining

About 5 µl of 2.5% sodium fluorescein was applied to mouse eyes followed by flushing with PBS to highlight corneal epithelial defect. The slit lamp images of the highlighted area were photographed with Cobalt blue filter. The staining area was analyzed with ImageJ (National Institutes of Health, Bethesda MD).

### Measurement of CGRP level in the cornea

Endogenous CGRP was quantitated in the mouse cornea or aqueous humor samples using a commercial enzyme-linked immunosorbent assay (ELISA) kit (Novus Biologicals LLC., Centennial, CO), as per the manufacturer's instructions with a detection level of 9.38 pg/ml. The concentration of the CGRP in the samples was calculated by comparing the OD of the samples to the standard curve.

### Real-time PCR

We used the RNeasy® Micro kit (Qiagen, Valencia, CA, USA) to isolate total RNA and the SuperScript™ III kit (Thermo Fisher Scientific Inc, Grand Island NY) was used to reverse RNA to cDNA as per the manufacturer's instructions. LightCycler® 480 II System (Roche Applied Science GmbH, Germany) was used to perform the RT-PCR cycle. The same protocol was followed with requisite modifications for extracting RNA from mouse corneal tissues and cultured cells. Please see Supplementary Table 1 for the detailed list of the primers used for RT-PCR experiments. All assays were performed in duplicates, and data were normalized to the housekeeping gene (Glyceraldehyde 3-phosphate dehydrogenase, *GAPDH*) mRNA levels and analyzed using the comparative threshold cycle method.

### Human corneal epithelial cell culture and migration assay

Telomerase-immortalized human corneal epithelium cells (hCEC) were kindly provided by Dr. Pablo Argueso[76]. The cells were seeded in 12-well plates and cultured in keratinocyte serum-free medium (KSFM) supplemented with 5 ng/ml epidermal growth factor and 50 µg/ml bovine pituitary extract (Life Technologies Inc., New York, NY). To assess the CGRP on hCEC functions, the cells were starved from the growth factors overnight before proceeding to the different treatments. To evaluate the effect of CGRP on hCEC migration, a linear scratch was made using a 10-µl micropipette tip in a confluent monolayer of epithelial cells at the bottom of the 12-well plate as previously described[77]. Subsequently, the cells were incubated in the growth factor-free (GFF) medium with various concentrations of CGRP (10–1000 nM) for 24 h. The migration area was

determined by comparing captured images over a period of 18 h using ImageJ software.

### Immunocytochemistry of hCEC

The proliferative effect of CGRP on hCEC was assessed by Ki67 staining. Briefly, the hCEC were passaged over 12 mm cover glass at a cell density of $2 \times 10^3$ cells/cm$^2$. Subsequently, the cells were incubated in the GFF medium supplemented with various concentrations of CGRP for 24 h. The cells were fixed using 4% paraformaldehyde (PFA) for 10 min, followed by PBS washing and permeabilization using PBS with a low-concentration detergent solution for 15 min. The cells were then kept in a 5% bovine serum albumin (BSA) blocking buffer for 30 min. Finally, the cells were incubated with Ki67 antibody for 1 h. The cells were washed and mounted with Vectashield with 4′,6-diamidino-2-phenylindole (DAPI) (Vector Laboratories Inc., Newark CA) and examined under a confocal microscope. For each sample, analysis was done by counting the Ki67-positive cells in five different fields and reporting the average.

To assess the effect of CGRP on ERK signaling in CEC, the cells were cultured in a GFF medium supplemented with 1000 nM CGRP for up to 3 h. The cells were fixed, permeabilized, and stained as described above with ERK1/2 antibody. All antibodies used in this study are reported in Supplementary Table 2.

### Western blot

Corneal tissues or cells were homogenized in RIPA lysis and extraction buffer (Thermo Fisher Scientific Inc, Grand Island NY), and total protein concentration was calculated using Bicinchoninic acid (BCA) assay kit (Thermo Fisher Scientific Inc, Grand Island NY). The denaturing reducing loading dye was added to the samples (matched with an equal amount of protein) and heated at 95 °C for 10 min. The samples were subjected to SDS–polyacrylamide gel electrophoresis (SDS-PAGE, Thermo Fisher Scientific Inc, Grand Island NY) using NuPAGE 4 to 12% Bis-Tris protein gels (Invitrogen Inc., Waltham MA) before transferring electrophoretically to a nitrocellulose membrane. The membrane was blocked with 5% BSA for 1 h The primary antibodies diluted in blocking solution supplemented with 0.1% Tween-20 detergent were applied to the membranes and incubated overnight at 4 °C on a shaker. The membrane was washed thrice with PBST before incubation in horseradish peroxidase (HRP) conjugated appropriate secondary antibody for 1 h at room temperature. The protein bands were visualized on Odyssey M fluorescent scanning system (LI-COR Corp., Lincoln NB) on adding Super Signal West Pico (Thermo Fisher Scientific Inc, Grand Island NY). Finally, the images were analyzed with ImageJ. Please see Supplementary Table 2 for the full list of the antibodies.

### Corneal fibroblast cell activation

The mice corneal fibroblast cell line (MK/T1) were cultured in Dulbecco's modified eagle medium (DMEM, Thermo Fisher Scientific Inc, Grand Island NY) supplemented with 10% fetal bovine serum (FBS)[78–80]. Prior to activation, cells were starved by reducing FBS to 1% overnight. Subsequently, the cells were activated with 10 ng/mL murine recombinant transforming growth factor (TGF-β1, R&D Systems Inc., Minneapolis MN)[59] with or without CGRP for 24 h. The fibroblast cells were collected to evaluate alpha-smooth muscle actin (α-SMA) expression.

### Immunohistochemistry and histology of mouse eyes

The entire eyes were harvested from mice fixed in formalin and subsequently embedded in paraffin, sectioned, and stained with hematoxylin and eosin (H&E). The stained tissues were examined using bright-field microscopy. For immunohistochemistry (IHC), the sections were de-paraffinized and blocked in 2% BSA supplemented with 0.1% Triton X-100 for 1 h at room temperature. The slides were then incubated with Ki67, laminin, TGF-β1, or α-SMA antibodies for overnight at 4 degrees. After washing with PBST, the slides were incubated with the appropriate fluorescent labeled secondary antibodies (Supplementary Table 2) diluted in 1% BSA. The slides were washed three times, and the staining was mounted with

Vectashield with DAPI and examined using TCS-SP8 confocal microscopy (Leica Camera A-G, Wetzlar, Germany). The corneal whole mount was stained using zonula occludens (ZO)-1 and $Na^+$/$K^+$ ATPase antibody[18].

## Flow cytometry

For flow cytometric analysis, we prepared single-cell suspensions from the harvested corneas. Briefly, the corneas were digested in RPMI-1640 medium (Lonza Biosciences Inc., Walkersville, MD) supplemented with 4 mg/mL collagenase type IV (Sigma-Aldrich Inc., St. Louis, MO) and 2 mg/mL DNase I (Roche Corp., Basel, Switzerland) for 45 min at 37 °C. Subsequently, the cells were passed through a 70-μm cell strainer (Corning Inc., Corning, NY). The cells were stained with CD45, CD11b, Ly6G, MHC-II, iNOS, AnnexinV, PI, or CCR2 antibodies (Biolegend Inc., San Diego CA). The appropriate isotypes were utilized as controls for the antibodies. The stained samples were evaluated using an LSR II flow cytometer (BD Biosciences Inc., San Jose, CA). The generated data were analyzed using the Summit software (Dako Inc., Colorado, CO). The gating strategy is presented in Supplementary Fig. 7.

## Statistics and reproducibility

All experiments were repeated at least twice and all attempts at replication were successful. The quantification of corneal opacity and area of wound healing was assessed in a masked fashion. Unpaired two-tailed Student's T-test was used to assess the difference between the two groups. For continuous variables of three groups or more, we performed a one-way analysis of variance (ANOVA) test with a Tukey-adjusted pairwise comparison. All the data in this manuscript are presented as means ± SEM. The compared data were considered statistically significant when the p values were less than or equal to 0.05.

## Reporting summary

Further information on research design is available in the Nature Portfolio Reporting Summary linked to this article.

## Data availability

All of the uncropped images in western blotting were shown in Supplementary Fig. 8. The source data behind the graphs in the manuscript were shown in Supplementary Data. The other data generated during and/or analyzed during the current study are available upon request from the corresponding author.

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

## Acknowledgements

This work was supported by the NIH/NEI grant K08 EY031340 (J.Y.), the Department of Defense office of the Congressionally Directed Medical Research Programs Project VR220057 Award Number HT9425-23-1-0951 (J.Y.), and the Schepens Eye Research Institute core grant P30 EY003790. The authors wish to thank Dr. Pablo Argüeso, PhD, Tufts University School of Medicine, for providing the human corneal epithelial cell line, Dr. Ula V. Jurkunas, Schepens Eye Research Institute, for providing the human corneal endothelial cell line, and Dr. Sunil K. Chauhan, Schepens Eye Research Institute, for providing the mice corneal fibroblast cell line.

## Author contributions

A.A.Z., S.Z. and J.Y. designed research; A.A.Z., S.Z., E.E., S.N. and Z.L. performed research; A.A.Z., S.Z., E.E., S.N. and A.N. analyzed data; and A.A.Z., S.Z., R.B.S. and J.Y. wrote the paper.

## Competing interests

The authors declare no competing interests.
