## [Peer Review File · Communications Biology]

Reviewers' comments:

Reviewer #1 (Remarks to the Author):

This study by Zidan and Yin demonstrated that Calcitonin gene-related peptide (CGRP) plays a cyto-protective, pro-regenerative, anti-fibrotic, and anti-inflammatory role in corneal wound healing. CGRP is a 37-amino acid peptide, which is primarily localized to C and A δ sensory fibers, abundant in the cornea. They reported that injury caused a reduction of CGRP levels in the cornea and topical application of CGRP accelerated corneal epithelial wound closure, reduced corneal opacification, and prevented corneal edema. CGRP affected all three residential cells and reduced neutrophil infiltration, macrophage maturation, and the production of inflammatory cytokines in the cornea. The experiments are well-designed and executed and the data presented support an important role of CGRP in mediating corneal wound healing.

Major concerns:

The authors proposed that CGRP may have therapeutic potential, and yet did not discuss the potential adverse effects of CGRP on migraine headaches which originated in the trigeminal nerve.

The authors stated that CGRP preserves corneal endothelial density and function. Are there experimental or clinical data showing corneal injuries cause endothelial loss?

Neutrophils and, more importantly, macrophages (M2) are required for proper wound healing, the implication of CGRP-mediated reduction of innate immune cells and cytokine production should be discussed.

The authors showed that CGRP plays a positive role in all three residential cell types, an experiment demonstrating the expression of CGRP receptors in corneal epithelial, stromal fibroblasts, and endothelial cells is helpful in supporting their conclusion.

The author claimed that epithelium-stroma injury allows CGRP to penetrate to the stroma due to the breakdown of the BM (in humans the Bowman's membrane), and yet they reported that CGRP acts on corneal endothelial cells that are separated from the corneal stroma by a BM. How might CGRP penetrate Descemet's membrane to protect the corneal endothelial structure and function?

Figure 4H showed that Ki67-positive cells were detected in CGRP-treated but not the control wounded corneas on day 4. It would be hard to understand that there were no proliferating epithelial cells in the normal corneas during re-epithelialization. Whole corneal should be shown for Ki67-positive cells to indicate that authors were not cherry-picking the IHC results.

Minor concerns:

Figure 4A, arrows should be used to highlight the cells they describe in the Results section.

The source of corneal fibroblast cells (MK/T1) was not mentioned. Are these cell lines authenticated?

The statement of "CGRP treatment significantly reduced TNF- α -induced activation of Caspase 3 (Cas3) and Bax" may not be correct, the activation of these genes first and foremost was by post-translational modification. This should be discussed.

TNF- α was officially named TNF.

The numbers mentioned in lines 197-200 do not match that of Figure 7A. Fig. 7E&F presenting percentages of neutrophils and macrophages, is not accurately reflecting the function of the cells as the numbers of a specific cell type in a cornea are more appropriate.

Reviewer #2 (Remarks to the Author):

This is a novel study on the effects on CGRP on corneal injury and wound healing. The study provides new insights into the cellular and molecular effects of CGRP in the cornea and its potential therapeutic application. Below are a few comments for the authors.

- While the 2mm injury model worked well for this study, for the future a limbal to limbal injury model which creates more inflammation and scarring may be interesting to study.

- Treatment was continued for 14 after the injury while the epithelial wound closes within the first few days. It would be interesting to study only short term treatment (until epithelial closure) since after epithelial closure and re-establishment of the barrier, it is not clear how much CGRP can get across the epithelium to reach the stroma/endothelium (if any data is available, please share)

- For the future, instead of PBS, it may be better to use a scrambled peptide or some other irrelevant protein (e.g. albumin) at the same concentration as the control.

We would like to thank the authors for their valuable comments and insights that improved the manuscript significantly. We have addressed the reviewers' comments as follows:

Reviewer #1 (Remarks to the Author):

This study by Zidan and Yin demonstrated that Calcitonin gene-related peptide (CGRP) plays a cyto-protective, pro-regenerative, anti-fibrotic, and anti-inflammatory role in corneal wound healing. CGRP is a 37-amino acid peptide, which is primarily localized to C and A δ sensory fibers, abundant in the cornea. They reported that injury caused a reduction of CGRP levels in the cornea and topical application of CGRP accelerated corneal epithelial wound closure, reduced corneal opacification, and prevented corneal edema. CGRP affected all three residential cells and reduced neutrophil infiltration, macrophage maturation, and the production of inflammatory cytokines in the cornea. The experiments are well-designed and executed and the data presented support an important role of CGRP in mediating corneal wound healing.

Major concerns:

1. The authors proposed that CGRP may have therapeutic potential, and yet did not discuss the potential adverse effects of CGRP on migraine headaches which originated in the trigeminal nerve.
- A. We agree with the reviewer's comment on the potential adverse effects of CGRP administration on migraine headaches. We are unable to assess rodent migraine behavior in our lab, but we did evaluate ocular pain response after injury with and without CGRP treatment. Mice respond to noxious stimulus such as hypertonic (2M) NaCl eyedrops by wiping their eyes. The frequency of eye wipes (assessed in a masked fashion) therefore has been used as an indicator for ocular pain (1, 2). Consistent with prior reports (3, 4), we found increased ocular pain response after corneal injury in the PBS control group, when compared with naïve uninjured animals, up to 14 days post-injury. Interestingly, topical application of CGRP did not heighten pain response at any time point observed. Rather, it decreased the pain response at Day 3 post-injury, likely due to accelerated epithelial healing. We've added these data to Supplementary Figure S6 and added the following discussion (Line 308-317):

“It is worth noting that CGRP administration (intracerebroventricular, intrathecal, intravenous, intraperitoneal) leads to migraine-like phenotypes through central and peripheral sensitization (5). However, there has not been report on the induction or development of migraine after ophthalmic application of CGRP to our knowledge. In our current study, we found increased ocular pain response after corneal injury up to 14 days post-injury (*SI*

Appendix, Fig.S6), consistent with prior reports (3, 4). Interestingly, topical application of CGRP did not heighten pain response at any time point observed. Rather, it decreased the pain response at Day 3 post-injury, likely due to accelerated epithelial healing. Nevertheless, further elucidation is needed to understand the impact of ophthalmic application of CGRP on migraine headache.”

2. The authors stated that CGRP preserves corneal endothelial density and function. Are there experimental or clinical data showing corneal injuries cause endothelial loss?

A. Yes, indeed, previous clinical studies have discussed the impact of trauma on corneal endothelial cells. We have added to the results (Line 167,168) the following:

“Previous studies have shown that ocular trauma can induce corneal endothelial cells (CEnC) swelling and loss (6, 7)”

And to the discussion (Line 298-301) the following:

“Previous clinical studies have reported such impact of the trauma on the CEnC leading to their swelling and disruption with fibrin and leukocyte accumulation (7). Moreover, measurable decreases in CEnC density, up to 20%, were noted in patients with history of blunt ocular trauma (6, 8, 9).”

3. Neutrophils and, more importantly, macrophages (M2) are required for proper wound healing, the implication of CGRP-mediated reduction of innate immune cells and cytokine production should be discussed.

A. We agree with the reviewer’s comment. In light of this, we studied the effect of CGRP on CD45⁺ cells’ infiltration in cornea 24 hours and 3 days after the injury. We found that CGRP treatment reduced (by approximately 40%), but did not eliminate, CD45⁺ cells infiltration compared with PBS control group. The frequency of their infiltration is still significantly higher than the naïve corneas. This result implies that the innate immune system is still active, but likely dampened, in the CGRP-treated eyes. We have added this data to Figure 7A and incorporated in the results accordingly. (Line 189-197)

“On day 1 post-injury, there was a notable increase in the infiltration of CD45⁺ cells in the cornea in the injured PBS-treated controls (9.7±1.0% of all corneal cells) compared to uninjured naïve corneas (0.07±0.04%). This effect persisted up to day 3 post-injury (2.1±0.3% of all corneal cells). Treatment with topical CGRP resulted in a reduction of CD45⁺ cell infiltration, decreasing to 5.7±0.4% on day 1 and further to 1.2±0.1% on day 3 post-injury (Fig. 7A, B). However, it is important to note that despite an approximately 40% reduction in immune cell infiltration after injury, CGRP-treated corneas still had significantly higher CD45⁺ cell infiltration than the naïve eyes. This result suggests

that CGRP mitigates, rather than completely inhibits, the innate immune response, which is crucial for proper wound healing.”

4. The authors showed that CGRP plays a positive role in all three residential cell types, an experiment demonstrating the expression of CGRP receptors in corneal epithelial, stromal fibroblasts, and endothelial cells is helpful in supporting their conclusion.
- A. We agree with the reviewer’s comment. We have added the data for receptors expression in the human corneal epithelial cells (hCEC), mouse corneal fibroblasts (mfibroblast) and human corneal endothelial cells (hCEnC) to the supplementary and incorporated it in the results (*SI Appendix, Fig.S3*), Line 132-133, and Line 152-153.

5. The author claimed that epithelium-stroma injury allows CGRP to penetrate to the stroma due to the breakdown of the BM (in humans the Bowman’s membrane), and yet they reported that CGRP acts on corneal endothelial cells that are separated from the corneal stroma by a BM. How might CGRP penetrate Descemet’s membrane to protect the corneal endothelial structure and function?

- A. We thank the reviewers for this comment. To answer this question, we analyzed CGRP concentration in the aqueous humor 30-minutes after CGRP administration. We did not observe any increase in the CGRP concentration in naïve mice or on Day 1 or Day 7 post-injury following CGRP administration. This indicates that topical application of CGRP does not penetrate through the cornea into the aqueous humor. Interestingly, we noted an increase in the CGRP concentration in the aqueous humor on Day 1 post-injury. The source of CGRP in the aqueous humor and this increase could be linked to sensory nerve stimulation (10) in the ciliary body (11-13).

We added these data to the results (Line 180-186):

“Having observed positive effects of CGRP on CEnC, we sought to investigate whether topically applied CGRP could penetrate the corneal stroma and the Descemet’s membrane to

reach CEnC. We quantified CGRP concentration in the aqueous humor 30 minutes after CGRP administration. However, no change in CGRP concentration was observed in naïve mice or on day 1 or day 7 post-injury following CGRP administration (*SI Appendix, Fig. S5*). This implies that the effect of CGRP on CEnC observed *in vivo* is likely via modulating the microenvironment (such as reducing tissue inflammation) rather than exerting a direct effect on CEnC.”

And discussion (Line 245-248)

“Interestingly, we noted an increase in CGRP concentration in the aqueous humor on day 1 after injury, followed by a decline on day 7 (*SI Appendix, Fig.S5*). The source of CGRP in the aqueous humor and this increase could be potentially linked to sensory nerve stimulation (10) in the ciliary body (11-13) and warrants further investigation.”

This suggests a lack of direct effect of CGRP on the CEnC. Accordingly, we have removed the *in vitro* data assessing the effects of CGRP on cultured CEnC from the manuscript. We highlighted the possible mechanism for the CEnC improvement in the discussion. (Line 308-311)

“Interestingly, topical application of CGRP does not change its concentration in the aqueous humor, whether in naïve or injured eyes. Therefore, the observed CEnC cyto-protection by CGRP is likely indirect and via its ability to dampen inflammatory response, which has deleterious effects on the CEnC (14-17).”

6. Figure 4H showed that Ki67-positive cells were detected in CGRP-treated but not the control wounded corneas on day 4. It would be hard to understand that there were no proliferating epithelial cells in the normal corneas during re-epithelialization. Whole corneal should be shown for Ki67-positive cells to indicate that authors were not cherry-picking the IHC results.
- A. We have replaced the Ki-67 images with lower magnification images to better highlight the distribution of the Ki-67 positive cells.

Minor concerns:

1. Figure 4A, arrows should be used to highlight the cells they describe in the Results section. The source of corneal fibroblast cells (MK/T1) was not mentioned. Are these cell lines authenticated?
 - A. Yes, these cell lines have been characterized (18) and used in previous publications(19, 20). We have added to the methods the reference characterizing the cell line.

2. The statement of “CGRP treatment significantly reduced TNF- α -induced activation of Caspase 3 (Cas3) and Bax” may not be correct, the activation of these genes first and foremost was by post-translational modification. This should be discussed. TNF- α was officially named TNF.
 - A. Thanks for the comment. Following Comment 5 above, we have removed this data from the manuscript.

3. The numbers mentioned in lines 197-200 do not match that of Figure 7A.
 - A. Thanks for the comment. Figure 7A is a representative one point from the 6 data points presented in Figure 7B. The number mentioned in 197 represents the mean value for all data points.

4. Fig. 7E&F presenting percentages of neutrophils and macrophages, is not accurately reflecting the function of the cells, as the numbers of a specific cell type in a cornea are more appropriate.
 - A. Thanks for the comment. We added the absolute number in the results as follows: (Line 208-213)

“Topical CGRP treatment significantly reduced the frequency of the infiltrating neutrophils to $0.6\pm 0.1\%$ (966.3 ± 246.4 cells/cornea) compared to PBS-treated controls $1.6\pm 0.1\%$ (3081 ± 505.2 cells/cornea) (Fig. 7E&F). Interestingly, the frequency of macrophages was comparable in the two groups, with $0.38\pm 0.02\%$ (301.5 ± 122.5 cells/cornea) in the PBS untreated group and $0.30\pm 0.1\%$ in the CGRP (261 ± 175 cells/cornea) (Fig. 7E&F).”

Reviewer #2 (Remarks to the Author):

This is a novel study on the effects on CGRP on corneal injury and wound healing. The study provides new insights into the cellular and molecular effects of CGRP in the cornea and its potential therapeutic application. Below are a few comments for the authors.

1. While the 2mm injury model worked well for this study, for the future a limbal-to-limbal injury model which creates more inflammation and scarring may be interesting to study.
- A. We agree with the reviewer wholeheartedly and we are currently assessing CGRP's effect on a limbal stem cell deficiency model.
2. Treatment was continued for 14 after the injury while the epithelial wound closes within the first few days. It would be interesting to study only short-term treatment (until epithelial closure) since after epithelial closure and re-establishment of the barrier, it is not clear how much CGRP can get across the epithelium to reach the stroma/endothelium (if any data is available, please share)
- A. We agree with the reviewer's comment. To address this comment, we have conducted another experiment where CGRP was applied only for 5 days. We found that CGRP-treated group exhibited a significantly lower opacity score compared to the PBS-treated group. However, this short course of CGRP treatment failed to improve corneal edema (as compared to the 14-day treatment). This suggest that the effect of CGRP on epithelial close and corneal opacity can be achieved with a short treatment course, whereas its therapeutic effect on suppression of inflammation and thus protection of corneal endothelial cells requires continuous and longer treatment course. We added these data to supplementary and to Line 124-131 as follows:

“To determine the short-term effect of CGRP on injury repair, we applied CGRP for 5 days only until the epithelial defect closed. We found that CGRP-treated group exhibited a significantly lower opacity score compared to the PBS-treated group. However, this short course of CGRP treatment failed to improve corneal edema, as compared to the 14-day treatment (*SI Appendix*, Fig.S2). This suggest that the effect of CGRP on epithelial close and corneal opacity can be achieved with a short treatment course, whereas its therapeutic effect on suppression of inflammation and thus protection of corneal endothelial cells requires continuous and longer treatment course”.

Regarding the penetration of CGRP into the cornea, we showed in Comment 5 to Reviewer 1 above that the topically applied CGRP does not increase CGRP concentration in the aqueous humor, suggesting that it does not penetrate through the entire corneal stroma and Descemet's membrane. We have thus removed sections on the direct in vitro effect of CGRP on cultured CEnC and revised the results and discussions accordingly.

3. For the future, instead of PBS, it may be better to use a scrambled peptide or some other irrelevant protein (e.g. albumin) at the same concentration as the control.
- A. We agree with the reviewer's comment. In other experiments on CGRP, we have used albumin instead of PBS as a control and observed similar results.

REFERENCES:

1. Farazifard R, Safarpour F, Sheibani V, Javan M. Eye-wiping test: a sensitive animal model for acute trigeminal pain studies. *Brain Res Brain Res Protoc.* 2005;16(1-3):44-9.
2. Nazeri M, Zarei MR, Pourzare AR, Ghahregh-Chahi HR, Abareghi F, Shabani M. Evidence of Altered Trigeminal Nociception in an Animal Model of Fibromyalgia. *Pain Med.* 2018;19(2):328-35.
3. Joubert F, Acosta MDC, Gallar J, Fakhri D, Sahel JA, Baudouin C, et al. Effects of corneal injury on ciliary nerve fibre activity and corneal nociception in mice: A behavioural and electrophysiological study. *Eur J Pain.* 2019;23(3):589-602.
4. Hegarty DM, Hermes SM, Morgan MM, Aicher SA. Acute hyperalgesia and delayed dry eye after corneal abrasion injury. *Pain Rep.* 2018;3(4):e664.
5. Wattiez AS, Wang M, Russo AF. CGRP in Animal Models of Migraine. *Handb Exp Pharmacol.* 2019;255:85-107.
6. Slingsby JG, Forstot SL. Effect of blunt trauma on the corneal endothelium. *Arch Ophthalmol.* 1981;99(6):1041-3.
7. Cibis GW, Weingeist TA, Krachmer JH. Traumatic corneal endothelial rings. *Arch Ophthalmol.* 1978;96(3):485-8.
8. Kim JH, Kim SK, Han SB, Lee SJ, Kim M. A case of traumatic corneal stromal edema with decreased endothelial cell density. *Int Med Case Rep J.* 2015;8:133-5.
9. Vohra V, Chawla H. Corneal endothelial decompensation due to airbag injury. *Int Ophthalmol.* 2018;38(5):2171-4.
10. Wahlestedt C, Beding B, Ekman R, Oksala O, Stjernschantz J, Håkanson R. Calcitonin gene-related peptide in the eye: release by sensory nerve stimulation and effects associated with neurogenic inflammation. *Regul Pept.* 1986;16(2):107-15.
11. Terenghi G, Polak JM, Ghatei MA, Mulderry PK, Butler JM, Unger WG, Bloom SR. Distribution and origin of calcitonin gene-related peptide (CGRP) immunoreactivity in the sensory innervation of the mammalian eye. *J Comp Neurol.* 1985;233(4):506-16.
12. Tamm ER, Flügel-Koch C, Mayer B, Lütjen-Drecoll E. Nerve cells in the human ciliary muscle: ultrastructural and immunocytochemical characterization. *Invest Ophthalmol Vis Sci.* 1995;36(2):414-26.
13. Seifert P, Stuppi S, Spitznas M, Weihe E. Differential distribution of neuronal markers and neuropeptides in the human lacrimal gland. *Graefes Arch Clin Exp Ophthalmol.* 1996;34(4):232-40.
14. Downie LE, Choi J, Lim JK, Chinnery HR. Longitudinal Changes to Tight Junction Expression and Endothelial Cell Integrity in a Mouse Model of Sterile Corneal Inflammation. *Invest Ophthalmol Vis Sci.* 2016;57(7):3477-84.
15. Meng J, Xu K, Qin Y, Liu Y, Xu L, Qiao S, et al. Tumor Necrosis Factor-Alpha Disrupts Cx43-Mediated Corneal Endothelial Gap Junction Intercellular Communication. *Oxid Med Cell Longev.* 2022;2022:4824699.
16. Chalimeswamy A, Thanuja MY, Ranganath SH, Pandya K, Kompella UB, Srinivas SP. Oxidative Stress Induces a Breakdown of the Cytoskeleton and Tight Junctions of the Corneal Endothelial Cells. *J Ocul Pharmacol Ther.* 2022;38(1):74-84.

17. Wang Q, Wei C, Ma L, Wang X, Li L, Zhou Q, Shi W. Inflammatory cytokine TNF- α promotes corneal endothelium apoptosis via upregulating TIPE2 transcription during corneal graft rejection. *Graefes Arch Clin Exp Ophthalmol*. 2018;256(4):709-15.
18. Gendron RL, Liu CY, Paradis H, Adams LC, Kao WW. MK/T-1, an immortalized fibroblast cell line derived using cultures of mouse corneal stroma. *Mol Vis*. 2001;7:107-13.
19. Biswas PS, Banerjee K, Kim B, Kinchington PR, Rouse BT. Role of inflammatory cytokine-induced cyclooxygenase 2 in the ocular immunopathologic disease herpetic stromal keratitis. *J Virol*. 2005;79(16):10589-600.
20. Biswas PS, Banerjee K, Kinchington PR, Rouse BT. Involvement of IL-6 in the paracrine production of VEGF in ocular HSV-1 infection. *Exp Eye Res*. 2006;82(1):46-54.

REVIEWERS' COMMENTS:

Reviewer #1 (Remarks to the Author):

The authors addressed well my comments and suggestions.

Reviewer #2 (Remarks to the Author):

The authors have addressed all my concerns, and my decision is Accept